# Updated Review on Clinically-Relevant Properties of Delafloxacin

**DOI:** 10.3390/antibiotics12081241

**Published:** 2023-07-28

**Authors:** Adrien Turban, François Guérin, Aurélien Dinh, Vincent Cattoir

**Affiliations:** 1Department of Bacteriology, University Hospital of Rennes, 2 Rue Henri Le Guilloux, 35000 Rennes, France; adrien.turban@chu-rennes.fr (A.T.); francois.guerin@chu-rennes.fr (F.G.); 2UMR_S 1230 BRM, Inserm/University of Rennes, 2 Avenue du Pr. Léon Bernard, 35000 Rennes, France; 3Infectious Diseases Department, University Hospital Raymond Poincaré, AP-HP, Paris Saclay, Versailles Saint Quentin University, 92380 Garches, France; aurelien.dinh@aphp.fr

**Keywords:** fluoroquinolones, levofloxacin, moxifloxacin, resistance, topoisomerases, ABSSSI, CAP

## Abstract

The extensive use of fluoroquinolones has been consequently accompanied by the emergence of bacterial resistance, which triggers the necessity to discover new compounds. Delafloxacin is a brand-new anionic non-zwitterionic fluoroquinolone with some structural particularities that give it attractive proprieties: high activity under acidic conditions, greater in vitro activity against Gram-positive bacteria—even those showing resistance to currently-used fluoroquinolones—and nearly equivalent affinity for both type-II topoisomerases (i.e., DNA gyrase and topoisomerase IV). During phases II and III clinical trials, delafloxacin showed non-inferiority compared to standard-of-care therapy in the treatment of acute bacterial skin and skin structure infections and community-acquired bacterial pneumonia, which resulted in its approval in 2017 by the Food and Drug Administration for indications. Thanks to its overall good tolerance, its broad-spectrum in vitro activity, and its ease of use, it could represent a promising molecule for the treatment of bacterial infections.

## 1. Introduction

Fluoroquinolones (FQs) are a family of antibiotics that have been extensively used in human medicine for more than 50 years [1]. Delafloxacin (DLX) is a brand-new anionic non-zwitterionic FQ that exhibits structural particularities, giving its unique chemical properties, including increased efficiency in acidic conditions, greater activity against Gram-positive pathogens, and dual targeting on both type II topoisomerases. Several clinical studies have evaluated its indication in the treatment of acute bacterial skin infections and skin structure infections (ABSSSIs), in community-acquired bacterial pneumonia (CABP), or in uncomplicated gonococcal infections [2,3,4,5]. It was first approved by the Food and Drugs Administration (FDA) in 2017 for the treatment of ABSSSIs and later authorized in CABP. Due to its broad-spectrum of activity, notably characterized by an increased efficiency against Gram-positive pathogens and regained efficacy against several FQ-resistant strains, DLX has already shown to be a useful tool in multiple other clinical situations. Indeed, favorable outcomes using DLX on multi-drug resistant strains have been described [6,7,8]. Since the use of FQ in human medicine has always been associated with the emergence of FQ-resistant strains, several studies have started to characterize and describe resistance patterns related to DLX, especially in *Staphylococcus aureus* [9,10,11], *Neisseria gonorrhoeae* [12], and more recently, in *Escherichia coli* [13,14].

The purpose of this review was to summarize the current scientific knowledge on this new compound: its particular structure-activity relationship and its impact on DLX properties, its in vitro activity, and its clinical efficacy and safety. This work also presents an overview of various quinolone resistance mechanisms and their impact on DLX. Finally, the review discusses the clinical significance of DLX within the FQ family and its potential future applications.

## 2. Structure-Activity Relationships and Mechanism of Action

Delafloxacin is the 1-(6-amino-3,5-difluoro-2-pyridinyl)-8-chloro-6-fluoro-7-(3-hydroxy-1-azétidinyl)-4-oxo-1,4-dihydro-3-quinolinecarboxylate (C_18_H_12_ClF_3_N_4_O_4_; molecular weight = 440.8 g/mol). The molecule differs from previous FQs due to several structural modifications articulated around the 4-quinolone core (Figure 1), which is common to all FQ molecules: (1) the presence of a large heteroaromatic substituent in position N-1, which increases the general activity of DLX and its steric hindrance, participate in restoring its activity on bacterial strains that show resistance to other FQs; (2) a chlorine in position R-8 enhancing its anti-Gram-positive and anti-anaerobic activity; and (3) the lack of a protonable group at R-7, resulting in a weak acidity that affects the distribution between ionized and non-ionized DLX under different pH values, leading to an increased activity under acidic conditions [11,15,16,17,18,19,20].

In neutral conditions (pH = 7.4), DLX prevails under anionic form (98.5%) (Figure 2). By contrast, when the environment is acidic (pH = 5.5), non-ionized DLX is dominant (50–62.5%) (Figure 2), which is unusual for FQs [16,21].

Used as a comparator, moxifloxacin (MXF) shows a completely different repartition, with a majority of the protonic form (89%) in an acidic environment and a predominance as a zwitterion under neutral conditions (92%) [16,20,21]. The absence of a protonable group at position R-7, as mentioned below, prevents DLX from existing under a zwitterionic form. That particular balance between ionized and non-ionized forms may account for the increased accumulation of DLX within bacterial cells. At pH 5.5, DLX exerts a 10-fold greater accumulation in *S. aureus* compared to pH 7.4. This property is correlated with better in vitro activity, characterized by a 4-to-5-fold decrease in MICs from neutral to acidic conditions. To date, MXF shows an opposite trend, characterized by a decrease in intra-bacterial accumulation and an increase in its MIC against *S. aureus* [21]. MIC decrease was also observed with DLX on *Escherichia coli* and *Klebsiella pneumoniae* and in *Stenotrophomas maltophilia* under acidic conditions [22,23,24]. The greater accumulation and activity of DLX under acidic conditions is probably explained by the unique distribution between its different molecular forms. As is generally acknowledged, non-ionized molecules diffuse more easily through cell membranes compared to ionized ones, which potentially explains why DLX, which predominates under uncharged forms in acidic environments, shows greater accumulation under low pH values. The neutral pH prevailing in bacterial cytoplasm favors the anionic form of DLX, making it less likely to diffuse through the membrane. Moreover, the acidic conditions can impact the expression of genes that are involved in influx and efflux mechanisms [21,22], leading to longer exposure of the targets to the antibiotic. FQ exerts its antibiotic pathway by targeting two enzymes: DNA gyrase and topoisomerase IV. DNA gyrase and topoisomerase IV are type-II topoisomerases that are essential for bacterial integrity, as they play a crucial role in replication, transcription, DNA repair, and recombination. Both enzymes are composed of two subunits A (GyrA for DNA gyrase and ParC for topoisomerase IV) and two subunits B (GyrB for DNA gyrase and ParE for topoisomerase IV). They both modulate local DNA topology by producing a transient double-strand break, allowing DNA strands to pass through each other. This transient breakage is stabilized by a “cleavage complex” composed of the topoisomerase and cleaved-DNA. DNA gyrase exerts a unique mechanism marked by its capacity to introduce negative supercoils to allow the progression of the replication fork. Topoisomerase IV is involved in decatenation of bacterial DNA thus allowing bacterial division [25,26,27]. That increased activity under acidic conditions may be a promising feature for the treatment of infections in acidic anatomic sites such as the urinary tract, skin, respiratory tract, or biofilm [18,24]. DLX also shows an equipotent dual-targeting of the two molecular targets of FQs in both Gram-negative and Gram-positive bacteria, even if it appears to target slightly more DNA gyrase in *S. aureus* [28]. That goes against the current knowledge of FQ, preferably targeting the DNA gyrase in Gram-negative bacteria and the topoisomerase IV in Gram-positive bacteria. This last property could explain the excellent in vitro activity of DLX against Gram-positive bacteria, as targeting DNA gyrase, which is located upstream of the replication complex, is more effective in inhibiting replication than targeting topoisomerase IV (located behind this complex) [29,30]. This particular dual targeting can probably decrease the risk of development of DLX-resistant mutants, which would require multiple simultaneous mutations on both targets [31,32]. This particular DLX structure also has an impact on its toxic profile. While the presence of a halogen (Cl) on the 4-quinolone core (R-8) increases the phototoxicity abilities of FQs [15], the association with a large heterocycle at N-1 decreases this risk of photoreaction. Furthermore, the anionic structure combined with the heteroaromatic substituent leads to reduced central nervous toxicity [33].

## 3. Pharmacokinetics and Pharmacodynamics

Delafloxacin is a bactericidal and concentration-dependent antibiotic. The best way to evaluate the FQ efficiency is by using the ratio between the area under the curve (AUC) and the MIC (AUC/MIC). Using the ratio C_max_/MIC (C_max_: maximal concentration) enables the assessment of the risk of resistance emergence to FQs [15]. To note, several chromatography methods are already available to monitor DLX plasmatic concentrations [34,35]. The studies of pharmacokinetics and pharmacodynamics (PK/PD) parameters of DLX have shown that the total exposure following a single 300 mg intravenous (IV) dose and a single 450 mg oral (PO, per os) dose is equivalent, allowing change in administration ways. The oral bioavailability of DLX is around 59% (Table 1) [36]. After a single IV dose (300 mg), the C_max_ is achieved at the end of the 1 h infusion (Table 1). Following a single oral dose (450 mg), the peak of serum concentration is reached between 1 to 2.5 h (Table 1) [37]. After multiple administrations (IV and PO), the C_max_ increases dose-proportionally, and the accumulation of the molecule is low [36,38]. Under a fed state, the total serum exposure of the molecule is not affected, even if the absorption is delayed from 1.5 to 3 h, which suggests that the drug can be administrated without consideration of the fed or fasted status [38]. No difference in PK/PD parameters was observed between men and women [38]. Total serum exposure was increased in elderly patients, likely due to the decrease in creatinine clearance in this population [38]. The “steady-state” is obtained after 3 days, and the volume of distribution (V_d_ = 35 L) reflects excellent distribution throughout the total body water. DLX shows a high binding to plasma proteins (84%) [36,39]. The mean half-life of DLX ranges from 10 h after IV infusion to 14 h after oral intake (Table 1) [36,37,38]. After IV administration, DLX is excreted into urines (ca. 66% of the dose) and in the feces (ca. 28%). Following PO administration, 50% of the dose is eliminated in the urine and 48% in the feces (Table 1).

Unmetabolized DLX (41%) and his glucuronide metabolite (20%) are found in the urine, whereas only DLX is excreted in the feces [31,39,40]. No inhibitory or inducible effects are observed on CYP450, BCRP, or P-gp [31,39,41]. To avoid chelation of the molecule, it is recommended not to administrate DLX 2 h before or 6 h after a medication involving multivalent cations (such as antacids containing magnesium or aluminum, iron, or zinc supplements) [31]. No data are available on potential interaction between DLX and rifampicin or with other particular drugs. Even in the absence of data, DLX should not be used during pregnancy and lactation or in the pediatric population since it belongs to the FQ class. A dosage adjustment should be considered for patients with severe renal insufficiency (creatinine clearance < 30 mL/min), as the dose should be reduced for those treated with the injectable form (oral administration remains unchanged) [42]. Its use is not recommended in patients with end-stage renal diseases. In patients with hepatic impairment, no dosage adjustment is necessary [43,44].

## 4. In Vitro Activity

DLX exhibits a broad spectrum of activity that includes Gram-positive and Gram-negative pathogens, atypical bacteria (*Legionella pneumophila*, *Mycoplasma pneumoniae*), and some anaerobic strains [45,46,47,48,49,50,51]. The European Committee on Antimicrobial Susceptibility Testing (EUCAST) has defined clinical breakpoints for the following bacterial species: *S. aureus* (≤0.016 mg/L and ≤0.25 mg/L in ABSSSIs), *Streptococcus pyogenes*, *Streptococcus dysgalactiae*, *Streptococcus agalactiae* and *Streptococcus anginosus* (≤0.03 mg/L) and *Escherichia coli* (≤0.125 mg/L) [41].

### 4.1. Activity against Gram-Positive Bacteria

DLX demonstrates excellent in vitro activity against Gram-positive bacteria. MIC_50_ and MIC_90_ for *S. aureus* are, respectively, ≤0.004 and 0.25 mg/L, with similar data on coagulase-negative staphylococci (CoNS) [45,49] (Table 2).

Against methicillin-resistant *S. aureus* (MRSA), DLX shows MIC_50_ of 0.06 to 0.25 mg/L and MIC_90_ of 0.25 to 1 mg/L, translating an activity at least 8-fold greater than levofloxacin (LVX) or MXF. Similar data have been obtained with methicillin-resistant CoNS (MR-CoNS) (Table 2). DLX shows at least an 8-fold higher activity than MXF against LVX-resistant strains of *S. aureus*, with MIC_90_ of 0.25 to 1 mg/L (Table 2) [10,11,48,55,56]. However, DLX seems to show lower activity against vancomycin-resistant *S. aureus* (VRSA) (MIC_90_, 4 mg/L; 7% of susceptibility), vancomycin-intermediate *S. aureus* (VISA) (MIC_90_,1 mg/L; 40% of susceptibility) and daptomycin-non-susceptible strains (MIC_90_, 1 mg/L; 38% of susceptibility) [56]. None of the linezolid-resistant (LR) *S. aureus* tested were susceptible to DLX (MIC_50_, 0.5 mg/L; 0% of susceptibility) [56]. By contrast, DLX shows low MIC_50_ (0.06 mg/L) and MIC_90_ (0.50 mg/L) against LR-*S. epidermidis* [57]. In comparison with vancomycin and daptomycin, DLX shows greater activity on biofilms induced by MRSA or methicillin-susceptible *S. aureus* (MSSA), probably because of the acidic pH within these bacterial structures [58,59]. Against *Enterococcus faecalis*, DLX displayed high in vitro activity (MIC_50_ and MIC_90_ at 0.06–0.12 and 1 mg/L, respectively), at least 8-fold greater than LVX and retained good efficiency on LVX-resistant strains (MIC_50_ at 1 mg/L) (Table 2) [45,52]. *Enterococcus faecium,* which usually shows high-level FQ resistance, is usually not susceptible to DLX (MIC_90_ > 4 mg/L) (Table 2). Nonetheless, DLX shows good activity against LVX-susceptible *E. faecium* strains (MIC_50_ and MIC_90_ at 0.12 and 1 mg/L, respectively) [28,45,55]. DLX shows a great activity when tested against *Streptococcus* spp. (MIC_50_ and MIC_90_ at 0.016 and 0.03 mg/L, respectively). The data on *Streptococcus pneumoniae* display an excellent activity of this new FQ (MIC_50_ and MIC_90_ at 0.008 and 0.016 mg/L, respectively), representing an activity 16- and 64-fold greater than those obtained with MXF and LVX, respectively [45,47,53]. Against LVX-resistant *S. pneumoniae*, DLX retains activity (MIC_50_ and MIC_90_ at 0.12 and 0.5 mg/L, respectively), showing higher efficiency than MXF (Table 2) [47,55,60]. However, in a series of *S. pneumoniae* strains isolated from patients with cancer, DLX exhibits lower activity (55% of susceptibility) than LVX (95% of susceptibility) [61].

### 4.2. Activity against Gram-Negative Bacteria

Against *Enterobacterales*, DLX shows similar activity to LVX and ciprofloxacin (CIP), with very good activity on *E. coli* and *K. pneumoniae* (Table 3) [61]. Overall susceptibility of *Enterobacterales* is lower when tested on strains that produce an extended-spectrum β-lactamase (ESBL) with MICs at least 32-fold higher (Table 3) [45,49].

Overall, data available show similar activity compared to current FQs against Enterobacterales [45]. Against Pseudomonas aeruginosa, DLX (MIC_50_ and MIC_90_ at 0.25–0.5 mg/L and >4 mg/L, respectively) exhibits no gain of efficiency when compared to CIP (MIC_50_ and MIC_90_ at 0.25 and >4 mg/L, respectively) (Table 3) [45,52]. A series of 28 CIP-resistant strains shows that DLX only retains activity in 36% of cases (10/28) [62]. Against respiratory pathogens like Haemophilus influenzae or Moxarella catarrhalis, DLX shows greater activity than LVX (Table 3) [47]. DLX also has high in vitro activity against Neisseria gonorrhoeae (MIC_50_ and MIC_90_ at 0.06 mg/L and 0.125 mg/L, respectively) (Table 3), even against CIP-resistant strains [12]. Against other non-fermentative Gram-negative bacilli, DLX exhibits MIC_50_ of 4 mg/L on *Achromobacter* spp., 0.25 mg/L on *Burkholderia cepacia* and *Burkholderia multivorans*, 2 mg/L for *Burkholderia cenocepacia* [63]. Against S. maltophilia, DLX shows greater activity under acidic conditions in comparison to LVX. Indeed, MIC_50_ of DLX decreases from 8 mg/L (in neutral pH conditions) to 0.25 mg/L (at pH = 6.5), whereas LVX values do not change. It is worth noting that the bactericidal activity of DLX was not improved at lower pH [24].

### 4.3. Activity against Other Bacteria

DLX also shows high activity against anaerobes. For example, MIC_50_ and MIC_90_ for *Clostridium* spp. are 0.032 and 1 mg/L, respectively [50]. Against other Gram-positive anaerobic rods (i.e., *Cutibacterium acnes*, *Propionibacterium avidum*, *Actinomyces* spp.), DLX is highly active (MIC_50_ and MIC_90_ at 0.008 and 0.032 mg/L, respectively). Both *Prevotella* spp. and *Bacteroides fragilis* show low MICs (MIC_50_ and MIC_90_ at 0.016 and 0.5 mg/L, respectively), which translates to an improvement in activity at least 64-fold higher than LVX [50]. Data on mycobacteria are scarce, but DLX shows a lower activity compared to MXF or CIP against several species, such as *Mycobacterium avium* complex, *M. abcessus,* or *M. chelonae*, with MICs ranging from 8 to 16 mg/L [64]. Interestingly, on *M. fortuitum*, DLX shows similar data to CIP and MXF, with MIC_50_ at 0.25 and MIC_90_ ranging from 0.50 to 2 mg/L [64,65]. One series of 14 isolates of *Legionella pneumophila* shows MIC_90_ of 0.125 mg/L [28]. Against *M. pneumoniae*, DLX shows good activity (MIC_50_ and MIC_90_ at 0.25 and 0.5 mg/L, respectively) [46]. Against *Bacillus anthracis,* the activity of DLX (MIC_90,_ 0.004 mg/L) was 16-fold higher than that of CIP (MIC_90_, 0.06 mg/L) [66]. DLX was also active on several sexually-transmitted bacteria such as *Chlamydia* spp., *Mycoplasma hominis,* or *Ureaplasma* spp. [67,68,69,70]. Against *M. hominis*, DLX exerts MIC_90_ at 1 mg/L, translating an overall activity 16- to 32-fold higher than MXF and LVX, respectively. Against *Ureaplasma* spp., DLX shows higher activity on both *Ureaplasma parvum* and *Ureaplasma urealyticum*, compared to MXF and LVX (MIC_90_ at, respectively, 2, 16 and >32 mg/L for *U. parvum* and 4, 16 and >32 mg/L for *U. urealyticum*), even showing regained activity on LVX-resistant strains [70].

## 5. Mechanisms of Resistance to Delafloxacin

### 5.1. Main Mechanisms of FQ Resistance

Resistance to FQs represents a worldwide issue that has led to a large restriction on the use of those molecules. FQ resistance can be acquired as a result of two types of mechanisms: acquisition of chromosomal mutations (Figure 3) and/or plasmid-mediated (Figure 4) resistance genes. Note that the main mechanism of resistance among clinical isolates results from chromosomal mutations, even if acquired transferable resistance genes are described in Gram-negative bacteria [71,72].

#### 5.1.1. Target Gene Mutation

The target gene mutation is the main FQ resistance mechanism. Most of the time, it occurs in the conserved region known as the quinolone resistance-determining region (QRDR) and mostly affects *gyrA* (for Gram-negative bacteria) and *parC* (for Gram-positive bacteria) through mutations in serine (90%), glutamic acid or aspartic acid residues located at positions 83/87 and 80/84, respectively (*E. coli* numbering) [15,73]. Mutations in *gyrB* and *parE* or outside the QRDR are possible but unlikely to occur [74]. This mechanism is cumulative: one mutation leads to an 8- to 16-fold increase in MIC, whereas multiple mutations exert a 10- to 100-fold augmentation of MIC [15,71,74,75].

#### 5.1.2. Decrease in Intracellular Drug Concentration

Overexpression of efflux-pumps can be observed in both Gram-positive and Gram-negative bacteria, only showing differences in their structural organization. In Gram-negative bacteria, the protein system is composed of an efflux pump located in the cytoplasmic wall, an outer membrane porin, and a fusion protein in the periplasmic space, allowing a connection between the last two membrane features. It usually belongs to the RND (resistance-nodulation cell division) family, such as AcrAB-TolC in *E. coli*, *Salmonella* spp. or *E. cloacae* complex, OqxAB-TolC in *K. pneumoniae*, or MexAB-OprM in *P. aeruginosa*. In Gram-positive bacteria, this system consists of a unique efflux pump and commonly belongs to the MFS (major facilitator superfamily) family, such as NorA in *S. aureus* and PmrA in *S. pneumoniae*. Moreover, two additional mechanisms leading to a decreasing intracellular drug concentration can be found specifically in Gram-negative bacteria: (1) alteration in the expression of major membrane porins (i.e., OmpF, OmpC, OmpD, OmpA, LamB, and Tsx) or in their structures, (2) disorganization of the outer membrane by modification of the LPS [15,71,75,76].

#### 5.1.3. Plasmid-Mediated Quinolone Resistance (PMQR)

Qnr is a plasmid-mediated protein responsible for target protection. About 100 variants have been described and are distributed into five types: QnrA, QnrB, QnrC, QnrD, and QnrS. Those proteins are capable of interacting with DNA gyrase or topoisomerase IV, thus disturbing interaction between FQs and their molecular targets [15,71,75,77]. Drug inactivation is mediated by an AAC(6′)-Ib-cr, a variant of an acetyltransferase that inactivates aminoglycosides. It acetylates the C7 unsubstituted nitrogen of the piperazine core found in CIP and norfloxacin (NFX), decreasing their antibiotic activity [15,71,75]. The last plasmid-mediated mechanism involves efflux pumps QepA and OqxAB. The first one belongs to the MFS family and only affects hydrophobic FQs such as CIP and NFX. OqxAB belongs to the RND family and shows a wide antibiotics spectrum (NFX, CIP, tetracycline, chloramphenicol, trimethoprim) [15,71,75].

### 5.2. Resistance to DLX

In contrast to other FQs, which show preferential targeting of DNA gyrase or topoisomerase, the dual targeting by DLX of both molecular targets can probably decrease the probability of the emergence of DLX-resistant mutants, which would require simultaneous multiple mutations in both targets [11,48,78,79,80]. The frequency of DLX-resistant MRSA selection in vitro is low (2 × 10^−9^ to <9.5 ×10^−11^) [11]. The mutant prevention concentration (MPC) in *S. aureus* with no mutation in QRDR was 8- to 32-fold lower (0.03 mg/L) than those obtained with MXF (0,25 mg/L) and LVX (1 mg/L), respectively [11,51,81]. Similar data were found for *S. pneumoniae*, *H. influenzae,* and *M. catarrhalis* [82]. *In vitro*-acquired resistance to DLX seems to happen when multiple simultaneous mutations appear on the two targets [9], as a single simultaneous mutation on both DNA gyrase and topoisomerase IV only leads to a slight MIC increase [9,37,48,83]. Current data show that S84L in *gyrA* and S80Y/F in *parC* are the most frequent mutations observed in DLX-resistant *S. aureus* strains. However, those mutations are also observed in DLX-susceptible *S. aureus* strains resistant to LVX and CIP. This suggests that those specific mutations are not sufficient to cause DLX resistance [10]. Indeed, higher MIC values for *S. aureus* appear with at least two mutations in both *gyrA* (mostly E88K and S84L) and *parC* (mostly E84G and S80Y/F) [2,3,9,10,48]. Remarkably, position 84 in *parC* seems to be a key point in reduced susceptibility to DLX since this mutation leads to high-level resistance [10]. In a study on *S. aureus*, the identification of new mutations in the *gyrA, gyrB, parC,* and *parE* genes, associated with a wide range of DLX MIC (0.5–32 mg/L), may also play a role in DLX-resistance. Moreover, the plasmid-encoded *qacC* gene (coding for an MFS-type efflux pump) has been recently described and may participate in DLX resistance in *S. aureus* [10]. DLX shows great bactericidal activity against *S. aureus* strains, showing zero, three, or four mutation(s) in the QRDR [83]. Concerning other mechanisms of resistance known in FQs, DLX seems to be a poor substrate for efflux transporters (NorA, NorB, NorC) in *S. aureus,* NorM in *N. gonorrhoeae* or in *Burkholderia pseudomallei* [11,12,84]. In Gram-negative bacteria, although resistance determinants have not been fully described in the literature, data available show that DLX-resistant *E. coli* strains contained at least three mutations in the QRDR [14,37]. Furthermore, a recent study exhibits the potential role of the AcrAB-TolC efflux pump in persistence and resistance to DLX treatment [13]. However, it is important to note that some strains of *N. gonorrhoeae* show new mutations in *gyrA, parC,* and *parE* genes and in the multidrug-resistance efflux (MtrC-MtrD-MtrE) or in NorM in two strains with MIC of 1 mg/L [12].

## 6. In Vivo Efficacy

### 6.1. Animal Models

The efficacy and various PK/PD parameters of DLX were evaluated in different murine models: lung infection models [85,86,87] and a renal abscess model [88]. In the murine lung infection model where *S. pneumoniae*, *S. aureus,* and *K. pneumoniae* were used, DLX demonstrated high in vitro and in vivo activity. It exhibited significant lung penetration, reflected by higher concentrations in the epithelial lining fluid (ELF) than those in the plasma [85]. Traces of DLX can also be found in rat bone after multiple weeks of IV and PO treatment, although no clinical data are available [37]. As with other FQs, the best way to predict treatment efficacy is by monitoring the AUC/MIC ratio [85,87]. The AUC/MIC ratio required to achieve bacteriostasis in both *S. aureus* and *S. pneumoniae* was 50- to 100-fold lower than those obtained with current FQs [86]. In a neutropenic lung infection model in mice involving *S. pneumoniae, S. aureus* (including MRSA), and *K. pneumoniae*, DLX shows greater activity than LVX used as a comparator [86]. In a murine renal abscess model, DLX shows a significant decrease in the bacterial load compared to MXF [88].

### 6.2. Phase II Clinical Trials

Two randomized, multicenter, double-blind phase II clinical trials were carried out to evaluate the efficacy and safety of DLX in the treatment of ABSSSIs [89,90]. A total of 406 patients were included; 150 in the first study that compared two dosings of DLX (300 mg and 450 mg) to tigecycline (100 mg) [89], and 256 patients in the second one that compared DLX (300 mg), vancomycin (15 mg/kg) and linezolid (600 mg) [90]. All treatments were administrated using IV infusion. A summary of all the characteristics of these studies can befound in Table 4.

Both groups of DLX dosing showed non-inferiority to tigecycline following clinical cure rate and microbiological eradication at test-of-cure visit (TOC) [89]. When compared to vancomycin and linezolid, DLX showed greater efficiency than vancomycin, as assessed by the investigators’ evaluation of clinical response, and showed similar clinical efficacy to linezolid. Concerning microbiological outcomes, no difference was observed in all three groups [90].

### 6.3. Phase III Clinical Trials

Efficiency and safety of DLX were also studied in 4 phase III clinical trials. Two of these included patients with ABSSSIs [2,3]; one was conducted on patients presenting CABP [5], and one included patients showing non-complicated gonorrhea [4].

#### 6.3.1. ABSSSI Phase III Clinical Trials

Two randomized, multicenter, double-blind phase III clinical trials were conducted in patients suffering from ABSSSIs. A total of 1510 patients were included to compare the efficacy and safety of DLX against a combination of vancomycin (15 mg/kg/12 h, IV) and aztreonam (1–2 g/12 h, IV). The characteristics of these studies are described in Table 5. The first one included 660 patients and compared DLX (300 mg, IV) to the aforementioned combination. Both arms were similar following the primary efficacy endpoint defined by the FDA (78.2% vs. 80.9%, respectively) and EMA (52% vs. 50.5%, respectively) (See Table 6). The additional secondary endpoint, as defined by EMA, also demonstrated non-inferiority of DLX compared to the combination (Table 6) [2]. In the obese population, DLX seems to be more effective than vancomycin/aztreonam at late follow-up (LFU) using the EMA primary criteria [2]. *S. aureus* was predominant in both groups and proportions of MRSA were similar in the two populations [2]. As shown in the literature, DLX exhibited excellent in vitro activity against *S. aureus* (MIC_50_ and MIC_90_ at 0.008 and 0.25 mg/L, respectively) and MRSA (MIC_50_ and MIC_90_ at 0.12 and 0.25 mg/L, respectively). These in vitro activities were well correlated with excellent in vivo efficacy. Indeed, documented or presumed eradication of *S. aureus* was found in 98.3% of patients for both groups [2]. All MRSA strains were eradicated in the DLX arm versus 98.5%, and all LVX-resistant *S. aureus* were eradicated in the DLX group [2]. The overall documented or presumed eradication was similar in both arms (97.8% for DLX and 98.4% for the association) [2]. The second study included 850 patients and was similarly designed but included a PO relay in the DLX arm, with a change in the dosing at day 3 of inclusion (300 mg/12 h to 450 mg/12 h) [3]. Based on the FDA primary endpoint, *per os,* DLX showed non-inferiority to vancomycin/aztreonam (83.7% vs. 80.6%, respectively) (Table 6). In the intent-to-treat population, DLX showed non-inferiority following the EMA primary endpoint (57.8% and 59.7%, respectively) (Table 6). However, in the clinically-evaluable population, this non-inferiority could not be statistically proven (lower limit of CI < −10%), thus preventing the definition of strict non-inferiority of oral DLX compared to the combination of vancomycin/aztreonam, based on the EMA primary criteria (Table 6). Microbiological eradication rates were similar in both arms (97.8% and 97.6%, respectively), and in vitro data of activity against *S. aureus* and MRSA were consistent with those obtained in the first study [2,3]. Cumulative microbiological data on the two studies showed that out of 81 cases of infections caused by LVX-resistant *S. aureus*, bacterial eradication was achieved in 80 cases. The most frequent mutations on these strains were S84L in GyrA and S80Y in ParC. The clinical impact of these mutations was insignificant, as demonstrated by an eradication rate of 98.6% in *S. aureus* carrying these mutations [91].

#### 6.3.2. CABP Phase III Trial

One randomized, multicenter, double-blind phase III clinical trial was conducted to evaluate the efficacy and toxicity of DLX (300 mg/12 h, IV) versus MXF (400 mg/24 h, IV) [5]. In both arms, a PO relay was possible after six infusions. If an MRSA was found in the MXF group, a change to linezolid (600 mg /12 h, IV) was done, and blinding was maintained. A total of 859 patients were included in the studies. Efficacy was evaluated during a control-visit that occurred 96 h after the first dose to define the early clinical response (ECR), primary criteria defined by FDA, and classify patients as responders or non-responders to treatment [5]. Test-of-cure (TOC) was conducted 5–10 days after the end of treatment, and a FU visit was performed at 28 days [5]. All criteria are summarized in Table 7. DLX demonstrated non-inferiority compared to MXF in the ECR rate of responders (88.9% and 89%, respectively) (Table 7). The secondary efficacy endpoint, as defined in Table 7, was in favor of DLX, which showed better results (52.7% vs. 43%, respectively) (Table 7) [5]. Interestingly, DLX showed greater activity in specific populations, such as patients suffering from chronic obstructive pulmonary disease (COPD) or asthma (93.6% in the DLX group vs. 76.8% in the MXF arm). Regarding the microbiological aspect, no difference was observed between the two groups [5]. DLX showed a 16-fold greater activity than MXF against Gram-positive and fastidious Gram-negative bacteria in the microbiological intent-to-treat population [5]. 

#### 6.3.3. Uncomplicated Gonorrhea Phase III Trials

An open-label, multicenter, randomized phase III trial evaluates a single oral dose of DLX (900 mg) vs. a single intramuscular dose (250 mg) of ceftriaxone (CTX) in patients with uncomplicated gonorrhea [4]. The primary efficacy endpoint was a microbiological cure (Table 7) [4]. DLX did not demonstrate non-inferiority compared to ceftriaxone, translating by microbiological cure rate significantly lower than those of the CTX arm (85.1% and 91%, respectively) (Table 7) [4]. Despite DLX’s excellent in vitro activity against *N. gonorrhoeae* [12], the molecule did not show non-inferiority to CTX in this clinical trial [4].

## 7. Safety

Clinical trials have shown that DLX is generally well-tolerated and that the occurrence of adverse events (AEs) was not statistically different between DLX and comparator arms [5,33,91,92]. The majority of AEs were mild to moderate in intensity. During phase III clinical trials, the most frequent AEs were gastrointestinal disorders (including vomiting, diarrhea, and nausea) along with headaches [33,91,93], which is consistent with previous studies [36,37,89,90]. Oral DLX did not show an increase in gastrointestinal AEs [33]. Pooled data of both phase III clinical trials on ABSSSIs show that fewer patients experienced AEs generally related to FQs in the DLX arms compared to the vancomycin/aztreonam group [33]. No difference was found in the occurrence of peripheral neuropathy related to the treatment. Additionally, no episodes of tendon rupture, phototoxicity, and convulsion were reported during phase III studies. Moreover, no QT prolongations were found either, which is consistent with previous data obtained demonstrating that even under supratherapeutic doses (900 mg), DLX does not cause significant QT interval modifications [94].

## 8. Conclusions

The higher activity of DLX in acidic conditions is interesting, as several infected anatomic sites, such as the respiratory or urinary tract, skin, or biofilms, often have low pH values. The dual targeting of both type-II topoisomerases in DLX appears to reduce the potential for the emergence of DLX-resistant mutants, as evidenced by the very low frequency of mutant selection in *S. aureus* [11]. Its increased activity against Gram-positive bacteria, especially on methicillin-resistant staphylococci, is an interesting property, as methicillin resistance is often related to FQ resistance in these species. It has also been shown that DLX retained an in vitro activity on LVX-resistant staphylococcal isolates, which holds promise for the treatment of certain infections caused by these species, like bone and joint infections. Indeed, some cases have already been reported describing the clinical efficacy of DLX [6,7,8]. Even if few data are available concerning the diffusion of DLX, it is well-established that FQs generally exhibit good diffusion into various tissues, notably into bones and joints [95,96], prostatic tissue [97,98,99], or lung tissue. In the lungs, DLX exhibits a high diffusion rate, similar to other FQs [85].

As part of the potential future aspect of DLX in human medicine, studying the diffusion of DLX into other tissues, in particular bones and joints, could be interesting, as the prevalence of methicillin-resistant staphylococci is important in that type of infection. A good diffusion into this tissue associated with excellent activity against both MRSA and MR-CoNS could make DLX a valuable treatment option in these infections. DLX may also hold promise as a therapeutic alternative for the treatment of urinary tract infections (UTIs), especially in men with prostatitis, as it is predominantly eliminated under its non-metabolized form via urine. DLX is, therefore, a promising antibiotic that exerts a large spectrum of activity (targeting both Gram-positive and Gram-negative bacteria) and interesting PK/PD properties. During clinical trials, patients were exposed to DLX for a short period of time (5 to 14 days), and DLX was generally well-tolerated, as the majority of AEs were gastrointestinal disorders and headaches [54]. Therefore, AEs related to its current use should be monitored, especially for those known to be associated with FQs (e.g., QT prolongation, hepatic or tendon disorders), particularly in case of prolonged treatments.

## Figures and Tables

**Figure 1 antibiotics-12-01241-f001:**
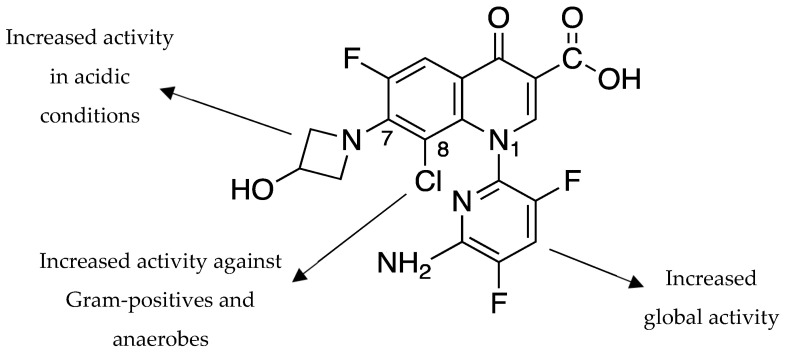
Structure-activity relationships (SAR) of delafloxacin.

**Figure 2 antibiotics-12-01241-f002:**
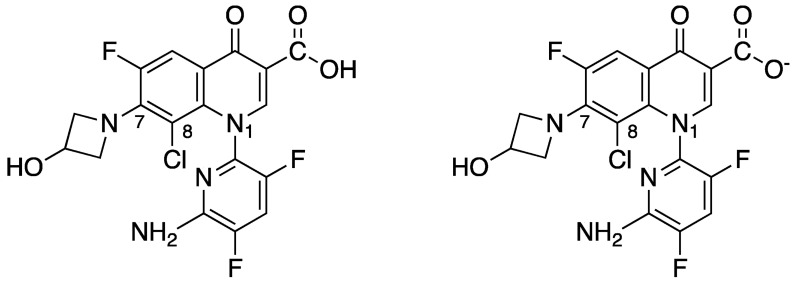
Representation of the non-ionized (**left**) and ionized anionic (**right**) forms of delafloxacin.

**Figure 3 antibiotics-12-01241-f003:**
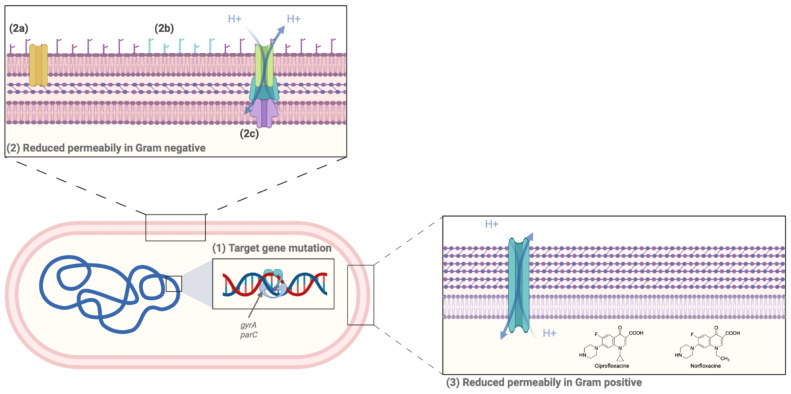
Chromosomal-encoded resistance mechanisms to FQs. (1) Target gene mutation occurs in QRDRs, mostly in gyrA and parC. (2a) Modification in membrane porins expression or in their structural features. (2b) Outer membrane disorganization (LPS modification). Both 2a and 2b are specific to Gram-negative bacteria. (2c) and (3) efflux pump overexpression (created with www.biorender.com, accessed on 20 June 2023).

**Figure 4 antibiotics-12-01241-f004:**
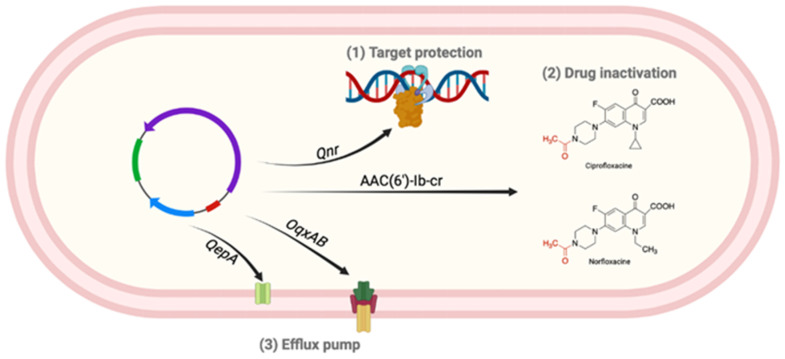
Plasmid-mediated resistance mechanisms to FQs. (1) Qnr proteins protect DNA gyrase and topoisomerase IV from interaction with FQs. (2) AAC(6′)-Ib-cr is able to acetylate unsubstituted piperazine core as found in ciprofloxacin and norfloxacin. (3) QepA efflux-pump only affects hydrophobic FQs such as ciprofloxacin and norfloxacin, whereas OqxAB exerts wide drug specificity (Created with www.biorender.com, accessed on 20 June 2023).

**Table 1 antibiotics-12-01241-t001:** Main pharmacokinetics parameters of delafloxacin.

Principal Pharmacokinetic Parameters of DLX		
T_max_	1 h (IV)	
	1–2.5 h (PO)	
V_d_	35 L	
Oral bioavailability	59%	
Protein binding	84%	
Half-life	IV	PO
	10 h	14 h
Elimination	66% (urines)	50% (urines)
	28% (feces)	48% (feces)

IV, Intravenous; PO, per os.

**Table 2 antibiotics-12-01241-t002:** In vitro activity of delafloxacin and comparators against clinical isolates of Gram-positive bacteria [10,11,45,47,48,49,51,52,53,54].

Organism/Antimicrobial Agent	MIC_50_ (mg/L)	MIC_90_ (mg/L)	MIC Range (mg/L)
*Staphylococcus aureus*			
Delafloxacin	≤0.004	0.25	≤0.004–8
Levofloxacin	0.25	>4	≤0.12–>4
Moxifloxacin	≤0.06	2	≤0.06–>4
Ciprofloxacin	64	>128	64–>128
MSSA			
Delafloxacin	≤0.004	0.008	≤0.004–4
Levofloxacin	0.25	0.25	≤0.12–>4
Moxifloxacin	≤0.06	≤0.06	≤0.06–>4
Ciprofloxacin	>128	>128	>128
MRSA			
Delafloxacin	0.06–0.25	0.25–1	
Levofloxacin	4	>4	
Moxifloxacin	2	>4	
Levofloxacin-susceptible *S. aureus*			
Delafloxacin	0.008	0.008	0.002–0.12
Levofloxacin-resistant *S. aureus*			
Delafloxacin	0.25	0.25–1	0.004–4
Moxifloxacin	2	8	
CoNS			
Delafloxacin	≤0.004	0.06	≤0.004–1
Levofloxacin	0.25	4	≤0.12–> 4
MR-CoNS			
Delafloxacin	0.06	0.5	≤0.004–2
Levofloxacin	4	>4	≤0.12–>4
Levofloxacin-susceptible CoNS			
Delafloxacin	≤0.004	0.06	≤0.004–1
Levofloxacin	0.25	4	≤0.12–> 4
Levofloxacin-resistant CoNS			
Delafloxacin	0.06	0.5	≤0.004–2
Levofloxacin	4	>4	≤0.12–>4
*Streptococcus* spp.			
Delafloxacin	0.016	0.03	
Levofloxacin	0.5	1	
Moxifloxacin	≤0.012	0.25	
*Viridans* group streptococci			
Delafloxacin	0.016	0.03	≤0.004–2
Levofloxacin	1	2	≤0.12–>4
Moxifloxacin	≤0.012	0.25	≤0.12–>4
*Streptococcus pyogenes*			
Delafloxacin	0.008	0.03	≤0.004–0.06
Levofloxacin	0.5	1	0.12–4
Moxifloxacin	≤0.012	0.25	≤0.012–2
*Streptococcus agalactiae*			
Delafloxacin	0.008–0.016	0.016–0.03	0.004–1
Levofloxacin	0.5	1	0.5–>4
Moxifloxacin	≤0.012	0.25	≤0.012–4
*Streptococcus dysgalactiae*			
Delafloxacin	0.008	0.016	≤0.004–0.12
Levofloxacin	0.5	1	0.12–>4
Moxifloxacin	≤0.012	0.25	≤0.012–2
*Streptococcus pneumoniae*			
Delafloxacin	0.008	0.016	≤0.004–0.5
Levofloxacin	1	1	0.5–>4
Moxifloxacin	≤0.012	0.25	≤0.12–>4
Levofloxacin-resistant *S. pneumoniae*			
Delafloxacin	0.12	0.5	0.016–1
Levofloxacin	>4	>4	>4
Moxifloxacin	2	4	0.25–>4
*Enterococcus faecalis*			
Delafloxacin	0.06–0.12	1	≤0.004–2
Levofloxacin	1	>4	0.25–>4
Levofloxacin-resistant *E. faecalis*			
Delafloxacin	1	2	0.06–2
*Enterococcus faecium*			
Delafloxacin	>4	>4	0.008–>4
Levofloxacin	>4	>4	0.5–4

CoNS, coagulase-negative staphylococci; MR-CoNS, methicillin-resistant CoNS; MRSA, methicillin-resistant; *S. aureus*; MSSA, methicillin-susceptible *S. aureus*.

**Table 3 antibiotics-12-01241-t003:** In vitro activity of delafloxacin and comparators on clinical strains of Gram-negative bacteria [11,45,47,48,49,51,52,53,54].

Organism/Antimicrobial Agent	MIC_50_ (mg/L)	MIC_90_ (mg/L)	MIC Range (mg/L)
*Enterobacterales*			
Delafloxacin	0.06	4	≤0.004–>4
Levofloxacin	≤0.12	>4	≤0.12–>4
Ciprofloxacin	≤0.03	>4	≤0.03–>4
*Escherichia coli*			
Delafloxacin	0.03	4	≤0.004–>4
Levofloxacin	≤0.12	>4	≤0.12–>4
Ciprofloxacin	≤0.03	>4	≤0.25–>4
Moxifloxacin	≤0.25	>4	≤0.03–>4
ESBL-producing *E. coli*			
Delafloxacin	2	>4	0.008–>4
Levofloxacin	>4	>4	≤0.12–>4
Ciprofloxacin	>4	>4	≤0.03–>4
*Klebsiella pneumoniae*			
Delafloxacin	0.06	>4	0.016–>4
Levofloxacin	≤0.12	>4	≤0.12–>4
Ciprofloxacin	≤0.03	>4	
ESBL-producing *K. pneumonia*			
Delafloxacin	4	>4	0.06–>4
Levofloxacin	>4	>4	≤0.12–>4
Ciprofloxacin	>4	>4	≤0.03–>4
*Klebsiella oxytoca*			
Delafloxacin	0.06	0.12	0.03–1
Levofloxacin	≤0.12	≤0.12	≤0.12–1
Ciprofloxacin	≤0.03	0.06	≤0.03–4
*Proteus mirabilis*			
Delafloxacin	0.06	2	0.016–>4
Levofloxacin	≤0.12	>4	≤0.12–>4
Ciprofloxacin	≤0.03	>4	≤0.03–>4
*Enterobacter* spp.			
Delafloxacin	0.06	1	≤0.004–>4
Levofloxacin	≤0.12	0.5	≤0.12–>4
Ciprofloxacin	≤0.03	0.25	≤0.03–>4
*Citrobacter* spp.			
Delafloxacin	0.06	2	0.008–>4
Levofloxacin	≤0.12	0.5	≤0.12–>4
Ciprofloxacin	≤0.03	0.5	≤0.03–>4
*Proteus* spp.			
Delafloxacin	0.12	4	0.008–>4
Levofloxacin	≤0.12	>4	≤0.12–>4
Ciprofloxacin	≤0.03	>4	≤0.03–>4
*Serratia* spp.			
Delafloxacin	1	2	0.03–>4
Levofloxacin	≤0.12	1	≤0.12–>4
Ciprofloxacin	0.12	1	≤0.03–>4
*Pseudomonas aeruginosa*			
Delafloxacin	0.25–0.5	>4	0.016–>4
Levofloxacin	0.5	>4	≤0.12–>4
Ciprofloxacin	0.25	>4	≤0.03–>4
*Haemophilus influenzae*			
Delafloxacin	≤0.001	0.004	≤0.001–0.25
Levofloxacin	0.016	0.016	0.004–>2
Ciprofloxacin	0.016	0.03	0.008–>2
*Moraxella catarrhalis*			
Delafloxacin	0.008	0.008	0.004–0.016
Levofloxacin	0.06	0.06	0.03–0.12
Ciprofloxacin	0.03	0.06	0.016–0.06
*Acinetobacter baumannii*			
Delafloxacin	2	>4	0.015–>4
Levofloxacin	>4	>4	≤0.012–>4
Ciprofloxacin	>4	>4	0.06–>4
*Neisseria gonorrhoeae*			
Delafloxacin	0.06	0.125	≤0.001–0.25
Ciprofloxacin	4	16	0.004–≥16

ESBL, extended-spectrum β-lactamase.

**Table 4 antibiotics-12-01241-t004:** Summary table of clinical and microbiological efficacy of delafloxacin during phase II clinical trials in patients with ABSSSIs.

Phase II Studies in ABSSIs								
Studies	Design	Population Size	Delafloxacin Group	Comparators Group	Duration	Monitoring	Evaluation Criteria	Results
O’Riordan et al., 2015 [89]	Multicenter, randomized (1:1:1), double-blind, non-inferiority	150	300 mg (IV), 2/24 h or 450 mg (IV), 2/24 h	Tigecycline (IV) 100 mg on day 1, then 50 mg, 2/24 h	5–14 days	14–21 days after last dose (TOC)	Primary efficacy clinical endpoint: clinical response Microbiological endpoint of eradication: documented eradicated, presumed eradicated, documented persisted, presumed persisted, superinfection, new infection	Primary efficacy endpoint: rate of cure at TOC: DLX 300 mg (94.3%), DLX 450 mg (92.5%), and TGC (91.2%) No difference between the three groups Microbiological endpoint of eradication at FU: No difference between the three groups
Kingsley et al., 2016 [90]	Multicenter, randomized (1:1:1), double blind, non-inferiority	256	300 mg (IV), 2/24 h	Vancomycin 15 mg/kg (IV), 2/24 h or Linezolid 600 mg (IV), 2/24 h	5–14 days	5 days after inclusion 14 days after inclusion(FU)	Primary efficacy clinical endpoint: investigator assessment of clinical response at FU defined patient as “cure”, “improved”, “failure”, or “indeterminate” Microbiological endpoint of eradication: documented eradicated, presumed eradicated, documented persisted, presumed persisted, superinfection, new infection	Primary efficacy endpoint: rate of cure at FU DLX (70.4%), LZD (64.9%) and VNC (54.1%) DLX significantly greater than VNC No difference between DLX and LZD Microbiological endpoint of eradication at FU: No difference between the three groups

TOC: Test-of-cure; FU: follow-up; LZD: Linezolid; TGC, tigecycline; VNC: vancomycin.

**Table 5 antibiotics-12-01241-t005:** Summary table of characteristics of phase III clinical trials in patients with ABSSSIs.

Phase III Studies in ABSSIs							
Studies	Design	Population Size	Delafloxacin Group	Comparators Group	Duration	Monitoring	Evaluation Criteria
Pullman et al., 2017 [2]	Multicenter, randomized (1:1), double-blind, non-inferiority	660	300 mg (IV), 2/24 h	Vancomycin 15 mg/kg (IV), 2/24 h and aztreonam 1–2 g (stopped if no Gram-negatives found)	5–14 days	48–72 h after first dose Day 14 (FU) Days 21–28 (LFU)	Primary efficacy endpoint defined by FDA: objective response at 48–72 h Primary efficacy endpoint defined by EMA: investigator assessment of clinical response at the FU Secondary efficacy endpoint defined by EMA: investigator-assessed success at FU Microbiological endpoint of eradication at FU
O’Riordan et al., 2018 [3]	Multicenter, randomized (1:1), double-blind, non-inferiority	850	300 mg (IV), 2/24 h with relay by 450 mg (PO)	Vancomycin 15 mg/kg (IV), 2/24 h and aztreonam 1–2 g (stopped if no Gram-negatives found)	5–14 days	48–72 h after first dose Day 14 (FU) Days 21–28 (LFU)	Primary efficacy endpoint defined by FDA: objective response at 48–72 h Primary efficacy endpoint defined by EMA: investigator assessment of clinical response at the FU Secondary efficacy endpoint defined by EMA: investigator-assessed success at FU Microbiological endpoint of eradication at FU

FU: follow-up; LFU: late follow-up.

**Table 6 antibiotics-12-01241-t006:** Summary table of clinical outcomes during phase III clinical trials in patients with ABSSSIs.

Phase III Studies in ABSSIs						
Studies	Subgroup	DLX (Events/Total)	%	VNC + AZT (Events/Total)	%	Percentage Difference (CI 95%)
Pullman et al., 2017 [2]	ITT					
	Objective response at 48–72 h ^a^	259/331	78.2	266/329	80.9	–2.6 (–8.78, 3.57)
	Investigator-assessed cure at FU ^b^	172/331	52	166/329	50.5	1.5 (–6.11, 9.11)
	Investigator-assessed success at FU ^c^	270/331	81.6	274/329	83.3	–1.7 (–7.55, 4.13)
	Investigator-assessed cure at LFU	233/331	70.4	219/329	66.6	3.8 (–3.27, 10.89)
	Investigator-assessed success at LFU	265/331	80.1	267/329	81.2	–1.1 (–7.15, 4.97)
	CE					
	Objective response at 48–72 h ^a^	250/294	85	257/297	86.5	–1.5 (–7.20, 4.18)
	Investigator-assessed cure at FU ^b^	142/240	59.2	142/244	58.2	1.0 (–7.79, 9.71)
	Investigator-assessed success at FU ^c^	233/240	97.1	238/244	97.5	–0.5 (–3.75, 2.72)
	Investigator-assessed cure at LFU	208/245	84.9	201/244	82.4	2.5 (–4.08, 9.15)
	Investigator-assessed success at LFU	237/245	96.7	241/244	98.8	–2.1 (–5.24, 0.70)
O’Riordan et al., 2018 [3]	ITT					
	Objective response at 48–72 h ^a^	354/423	83.7	344/427	80.6	3.1 (−2.0, 8.3)
	Investigator-assessed cure at FU ^b^	244/423	57.7	255/27	59.7	−2.0 (−8.6, 4.6)
	Investigator-assessed success at FU ^c^	369/423	87.2	362/427	84.4	2.5 (−2.2, 7.2)
	Investigator-assessed cure at LFU	287/423	67.8	303/427	71.0	−3.1 (−9.3, 3.1)
	Investigator-assessed success at LFU	353/423	83.5	351/427	82.2	1.3 (−3.8, 6.3)
	CE					
	Objective response at 48–72 h ^a^	346/395	87.6	327/387	84.5	3.1 (1.8, 8.0)
	Investigator-assessed cure at FU ^b^	220/353	62.3	224/329	68.1	−5.8 (−12.9, 1.4)
	Investigator-assessed success at FU ^c^	340/353	96.3	319/329	97.0	−0.6 (−3.5, 2.2)
	Investigator-assessed cure at LFU	259/337	76.9	267/323	82.7	−5.8 (−11.9, 0.3)
	Investigator-assessed success at LFU	322/337	95.5	310/323	96.0	−0.4 (−3.7, 2.8)

^a^ Primary endpoint defined by FDA; ^b^ Primary endpoint defined by EMA; ^c^ Additional endpoint defined by EMA; AZT, aztreonam; DLX, delafloxacin; VNC.

**Table 7 antibiotics-12-01241-t007:** Summary table of clinical and microbiological efficacy of delafloxacin during phase III clinical trials in patients with CABP or uncomplicated gonorrhea.

**Phase III Studies**								
**Studies**	**Design**	**Population Size**	**DLX** **Group**	**Comparators** **Group**	**Duration**	**Monitoring**	**Evaluation Criteria**	**Results**
CABP, Horcajada et al., 2020 [5]	Multicenter, randomized (1:1), double-blind, non-inferiority	859	300 mg (IV), 2/24 h with PO relay (450 mg) after 6 doses	MXF 400 mg, 1/24 h PO relay was conducted. If MRSA was found, a relay for linezolid 600 mg IV was conducted	5–10 days	Early clinical response (ECR) at 96 h after first dose 5–10 days after end of treatment (TOC) Day 28 (FU)	Primary efficacy endpoint defined by FDA: early clinical response at 96 h: defined patients as responder to treatment or not Secondary efficacy endpoint defined by FDA: ECR in addition of improvement in vital signs Microbiological endpoint of eradication at TOC	Primary efficacy endpoint defined by FDA: DLX (88.9%), MXF (89%). No difference between both groups Secondary efficacy endpoint defined by FDA: DLX (52.7%), MXF (43%). Significant improvement in DLX groups Microbiological endpoint of eradication at TOC: No difference between both groups
Uncomplicated gonorrhea, Hook et al., 2019 [4]	Multicenter, randomized (2:1), open-label, non-inferiority	460	900 mg PO, single dose	CTX 250 mg, intramuscular, single dose	Single dose	Visit at 7 ± 3 days (TOC)	Primary efficacy endpoint defined by FDA: microbiological outcomes: cure or failure	Primary efficacy endpoint defined by FDA: DLX (85.1%), CTX (91%). DLX did not show non-inferiority versus CTX

CTX, ceftriaxone; DLX, delafloxacin; ECR, early clinical response; FU, follow-up; IV, intravenous; MRSA, methicillin-resistant *S. aureus*; MXF, moxifloxacin; PO, *per os*; TOC, test-of-cure.

## Data Availability

The data presented in this study are available in the article.

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
