# Peer review of "Updated Review on Clinically-Relevant Properties of Delafloxacin"

_antibiotics, 2023, doi:10.3390/antibiotics12081241_

Round 1
Reviewer 1 Report
Overall, the authors provide a reasonably detailed and thorough characterization of the bacterial antibiotic resistance. The text provides good evidence. The data are generally well presented and are worthy of being published. I do believe that the present manuscript is suitable for publication in this journal. I would suggest that the authors consider the following points as they revise their manuscript and continue their work in this important (antibiotic resistance) research area.
1. The introduction needs minor revision. They could check for recently published articles; please add a specific introduction that will perfectly match.
The objectives of this study are absent. After the introduction, one paragraph for the objectives statement is necessary.
2. Kindly add one paragraph of “future aspect”.
Minor editing of English language required
Author Response
Overall, the authors provide a reasonably detailed and thorough characterization of the bacterial a antibiotic resistance. The text provides good evidence. The data are generally well presented and are worthy of being published. I do believe that the present manuscript is suitable for publication in this journal. I would suggest that the authors consider the following points as they revise their manuscript and continue their work in this important (antibiotic resistance) research area.
Authors: Thank you for your interest and your time reviewing our work. Modifications mention in this response section were highlighted in the text to ease comprehension.
- The introduction needs minor revision. They could check for recently published articles; please add a specific introduction that will perfectly match.
Authors: Following modifications have been made and highlighted in the manuscript:
Lines 34-41 “Due to its broad-spectrum of activity, notably characterized by an increased efficiency against Gram-positive pathogens and regained efficacy against several FQ-resistant strains, DLX has already shown to be a useful tool in multiple other clinical situations. Indeed, favorable outcomes using DLX on multi-drug resistant strains have been described [6–8]. Since the use of FQ in human medicine has always been associated with emergence of FQ-resistant strains, several studies have started to characterize and describe resistance patterns related to DLX, especially in Staphylococcus aureus [9–11], Neisseria gonorrhoeae [12] and more recently, in Escherichia coli [13,14]. ” We also add one citation #14 Gulyás, D et al. 2023
- The objectives of this study are absent. After the introduction, one paragraph for the objectives statement is necessary.
Authors: Thank you for pointing out this limitation. Following modifications have been made and highlighted in the manuscript:
Lines 43-47 “its particular structure-activity relationship and its impact on DLX properties, its in vitro activity, and its clinical efficacy and safety. This work also presents an overview of various quinolone resistance mechanisms and their impact on DLX. Finally, the review discusses the clinical significance of DLX within the FQ family and its potential future applications.”
- Kindly add one paragraph of “future aspect”
Authors: Indeed, future aspect of delafloxacin are promising. As already mentioned in the conclusion, we added the following modification to point out the paragraph referring to the potential future use of delafloxacin.
Lines 443 “As part of potential future aspect of DLX in human medicine”
- Minor editing of English language required
Authors: English has been improved.
Reviewer 2 Report
It would be interesting if the authors supplement their review with more informations as:
What are the adverse effects?
Are there data about risk in pregnancy?
Are there data about CSF penetration?
What are major drug interactions?
Are there data about pediatic population?
Author Response
It would be interesting if the authors supplement their review with more informations as:
Author: Thank you for your interest and your time reviewing our work. Modifications mention in this response section have been highlighted in the text to ease comprehension.
- What are the adverse effects?
Authors: adverse effects are an important part of FQ antibiotics family. As mention in paragraph 7, delafloxacin is generally well tolerated and the most frequent AEs were gastrointestinal disorders and headaches. To resume that, we proposed the following sentence (lines 453-454): “[…] DLX was generally well tolerated, as the majority of AEs were gastrointestinal disorders and headaches”.
- Are there data about risk in pregnancy?
Authors: as mentioned in lines 145-147, data on pregnancy are lacking. However, since DLX is part of the FQ family, it is therefore not allowed in this population.
- Are there data about CSF penetration?
Authors: no data is available on CSF penetration.
- What are major drug interactions?
Authors: actually, no data are available on drug interaction involving DLX. As a member of FQ family, it should no be administrated with medication involving multivalent cations. Also, a mention of the lack of data was added in lines 144-145 “No data is available on potential interaction between DLX and rifampicin or with other particular drugs.”
- Are there data about pediatic population?
Authors: as mentioned in lines 145-147, data on pediatric population are lacking. However, since DLX is part of the FQ family, it is therefore not allowed in this population. To note, no clinical trial or case report have been reported in this particular population.